# Data Optimization for Industrial IoT-Based Recommendation Systems

Mykola Beshley [1,2,*], Olena Hordiichuk-Bublivska [1], Halyna Beshley [1,2] and Iryna Ivanochko [2,3]

1    Department of Telecommunications, Lviv Polytechnic National University, Bandera Str. 12,
     79013 Lviv, Ukraine
2    Department of Information Systems, Faculty of Management, Comenius University in Bratislava,
     82005 Bratislava, Slovakia
3    Department of Management and International Business, Lviv Polytechnic National University,
     79000 Lviv, Ukraine
*    Correspondence: mykola.i.beshlei@lpnu.ua

**Abstract:** The most common problems that arise when working with big data for intelligent production are analyzed in the article. The work of recommendation systems for finding the most relevant user information was considered. The features of the singular-value decomposition (SVD) and Funk SVD algorithms for reducing the dimensionality of data and providing quick recommendations were determined. An improvement of the Funk SVD algorithm using a smaller required amount of user data for analysis was proposed. According to the results of the experiments, the proposed modification improves the speed of data processing on average by 50–70% depending on the number of users and allows spending fewer computing resources. As follows, recommendations to users are provided in a shorter period and are more relevant. The faster calculation of modified Funk SVD to exchange the optimal parameters between nodes was proposed. It was determined that execution time can be reduced on average by 75% for using ten nodes exchanging the optimal decomposition parameter compared to using one. Using Spark technology for faster calculation on average by 20% compared to Hadoop was proposed. The architecture of the IIoT system was proposed, which uses a modified Funk SVD algorithm to optimize data on edge devices and monitors the effectiveness of providing recommendations using control centers and cloud resources.

**Keywords:** Industrial Internet of Things; Funk SVD; smart manufacturing; cloud manufacturing; recommendation systems; sparse matrix





## 1. Introduction

Data processing and optimization play an essential role in the operation of industrial systems. The use of cloud technologies contributes to storing a significant amount of data. Search and analysis of information allows monitoring of the state of Industrial Internet of Things (IIoT) systems and increases their efficiency. Collecting data from various subsystems allows early detection and prevention of problems. Recommendation systems in the IoT era provide personalized recommendations based on historical user datasets collected from IoT devices. The recommendation enables an efficient decision-making process by proposing relevant products, resources, and information. The expansion of the list of services provided by modern industrial systems has changed the approach to their design and use. Instead of centralized systems managed and maintained exclusively by humans, flexible and decentralized systems are now used. Industrial Internet of Things, cloud manufacturing (CMfg), and machine learning allow for creating a more efficient industrial system and providing users with quality services [1].

In the Industrial Internet of Things, data are collected from a significant number of terminal and edge devices. Smart manufacturing is considered a single system that includes the stages of production and sale to customers. In the Industrial Internet of

Things, information collected from various sources helps identify weak points in the production process and improve the quality of service to clients. In order to transform existing or create new smart productions, modern software and hardware tools should be used [2]. A large number of end devices that perform local data collection and analysis functions allow us to reduce the load on central control devices and better account for the specifics of specific subsystems [3]. Due to the methods of machine learning and artificial intelligence, industrial systems are able to learn from their own datasets and correct emergency situations before they occur [4]. This approach simplifies the process of troubleshooting, organizing the production process, and delivering goods directly to users. A change in system parameters can be considered as a signal of the need to modify its operation. The problem of smart industrial production is the need to constantly process a significant amount of data coming from various devices. For a quick result, the most important information should be determined, and information not necessary for the result should be discarded.

The distribution of production tasks among several devices reduces the load and speeds up the calculation process. The distributed architecture of industrial systems contributes to the more efficient operation of intelligent production and faster results. Decentralization of management functions allows more flexible and effective decision-making to solve local problems and account for the characteristics of individual users. The exchange of experience between end devices accelerates the training of local machine learning systems. Recommendation systems (RS) are used to work with data on the interaction of users and products [5]. RS not only establish statistical features and relationships among goods, services, and user behavior but also predict the preferences of individual users based on their previous actions. Due to this approach, it is possible to provide more personalized recommendations to users of industrial systems, take into account the demand for existing products, and adjust production parameters to achieve the greatest efficiency [6]. The involvement of end devices to collect information improves the quality of service provision and allows the creation of local learning models that take into account the specific characteristics of users. Special mathematical algorithms for data analysis in recommendation systems are used. One of them, Funk SVD, is considered in the paper, and its advantages for processing sparse matrices of large volume are given. The problem of processing big data and providing relevant recommendations to users in industrial systems is still relevant. We analyzed the Funk SVD algorithm and found it is effectively used for processing sparse big data matrices. However, no studies show its effectiveness when working in conditions where the load on the system can change dynamically, so the optimal parameters for big data processing should be selected accordingly. We analyzed existing solutions and proposed modifications of Funk SVD, which allowed us to speed up the calculation process by discarding redundant data about users. In addition, with high requirements for the accuracy and speed of data processing, we proposed the architecture of a recommender system with exchanging optimal calculation parameters that help form recommendations faster.

The main contributions and advantages of this study:

- The main research in the field of processing big data about users was analyzed, and the advantages of the Funk SVD algorithm were revealed;
- A modified Funk SVD was proposed, which works more efficiently in systems with high requirements for calculation speed because, when forming recommendations, data are not used about all users, but only a part, thus discarding information that does not significantly affect the accuracy of calculations;
- A second modification of Funk SVD was proposed, which allows exchanging optimal parameters between different nodes, which affects the effectiveness of providing recommendations;
- For calculating Funk SVD in the systems of the Industrial Internet of Things, the Spark distributed computing technology was proposed, and the improvement of the speed of calculations, especially of large datasets, was determined;

- Based on the research, the architecture of recommender systems in IIoT working with variable work requirements was proposed. In this way, it is possible to flexibly apply the proposed modifications to achieve the maximum efficiency of user data processing.

The remainder of the paper is organized as follows. In Section 2, we survey the most recent work on this topic and identify issues that still need to be addressed. The main features of smart production work and the necessary use of recommendation systems are determined in Section 3. A study of data processing from users of industrial systems using the Funk SVD algorithm is conducted, and a modification is proposed in Section 4. Section 5 concludes the paper and determines the directions for further research.

## 2. Related Work

In this section, we will review and analyze current work in this field. In [7], M. Maheswari and N.C Brintha looked at the benefits of smart manufacturing and the use of cloud technologies, the Internet of Things (IoT), and cyber security to improve it. C. Eyupoglu investigated the problem of processing big data using IoT and cloud computing [8]. In [9], M. Chen analyzed the improvement of information processing efficiency in geological information systems thanks to cloud technologies and the Internet of Things. J. Leng et al. investigated security issues in smart factories and proposed the integration of blockchain technology [10]. In [11], J. Behnke considered the challenges of creating a digital twin in a factory to improve production efficiency. J. Bai et al. [12] emphasized the importance of cloud manufacturing (CMfg) and proposed an automatic perfecting service information method of BOSS (Bills of Standard Manufacturing Service) for service standardized encapsulation. In [13], Y. Teng et al. explored a blockchain-enabled smart manufacturing system and offered load-balanced processing from users. S. Baer et al. considered improving the efficiency of flexible manufacturing systems using online scheduling [14]. In [15], C.-L. Liu et al. analyzed the job shop scheduling problem and offer its solution in a dynamic environment using deep learning and parallel training. L.M Gladence [16] investigated the improvement of smart manufacturing using cloud-based pluggable production processes and edge computing.

In [17], J. Yu et al. investigated collaborative filtering in recommendation systems and proposed an improved method to account for changes in user preferences. X. Jia and F. Liu analyzed the problems of big data processing in commercial systems and improved the real-time recommendation model [18]. F. Liu [19] investigated the effectiveness of recommendation systems using Spark technology. J. Jin et al. [20] proposed a recommendation system that improves the operation of the traffic management system by learning based on human behavior. D.V Bagul and S. Barve developed a recommendation system using latent Dirichlet allocation and the Jensen–Shannon distance [21]. In [22], S. Chen and J. Feng proposed fabric defects extraction using singular value decomposition. L.-Y. Gao et al. [23] investigated the use of the SVD algorithm for blind signal separation. In [24], J. Jiang et al. used singular value decomposition to identify geological targets. S. Guo and C. Li investigated the advantages of the Funk SVD algorithm for working with sparse data and proposed a new recommendation algorithm that combines Funk-SVD and K-means [25]. In [26], K. Birul proposed improving and simplifying Funk SVD using only one user's data when forming recommendations, proving that the accuracy of calculations does not deteriorate.

The paper's authors [27] developed a recommendation system based on machine learning that can be easily applied to almost any application. Unlike existing recommendation systems, the proposed system supports multiple types of interaction data with different ways of metadata through a multimodal fusion of different data representations, which contributes to a better result on the provision of recommendations by users.

The paper [28] has reviewed a significant amount of work on recommendation systems. The evolution of recommendation systems over the past 10 years has revealed an interconnectedness, in particular, with the growth of digital platforms for business; there has been a quantitative increase in various detailed studies of recommender systems. By

providing a comprehensive overview of recommender systems, this review of papers provides insight to the many researchers interested in recommender systems through an analysis of the various technologies and trends in the services to which recommender systems are applied. In [29], F. Zou et al. proposed modifying the Adam and RMSProp algorithms, which are the most influential adaptive stochastic algorithms. In [30–32], the authors developed original optimization methods that use the decomposition of singular values and the duality of uplink and downlink channels to optimize the weight vectors in order to form energy-efficient beamforming for a satellite-to-ground relay network. In [33], authors proposed secure communication in IoT networks by jointly optimizing the power allocation factors, beamforming vector, and phase shifts, where the confidential signal is sent by an active refracting reconfigurable intelligent surface-based transmitter, to solve a secrecy energy maximization problem.

Y. Xiaochen and L. Qicheng [34] proposed an improved FunkSVD algorithm based on the root-mean-square prop and graphics processing unit for faster data processing. Table 1 shows a comparison of the most recent studies along with advantages and disadvantages in the field of big data processing in recommender systems.

**Table 1.** Comparison of the most recent studies of big data processing in recommender systems.

| Study | Advantages | Disadvantages |
|:---:|:---:|:---:|
| [18] | Authors proposed an integrated online and offline real-time recommendation service providing a variety of analysis methods to realize data mining. | The possibilities of applying the proposed method and optimization of big data should be covered more. |
| [20] | Proposed a recommendation system that improves the operation of the traffic management system by learning based on human behavior. | No methods are given to optimize big data obtained from different users and adaptation to different workloads. |
| [25] | Investigated the advantages of the Funk SVD algorithm for working with sparse data and proposed a new recommendation algorithm that combines Funk-SVD and K-means. | Features of operation for variable load systems are not given. |
| [26] | Proposed to improve and simplify Funk SVD, using only one user's data when forming recommendations, proving that the accuracy of calculations does not deteriorate. | No comparison is given on how the effectiveness of providing recommendations changes when processing data from different numbers of users. |
| [27] | Developed a recommendation system based on machine learning that can be applied to almost any application. | There are no defined problems of different volume data processing and service requirements. |
| [34] | Offered a parallel GPU-based Funk SVD algorithm, demonstrating high accuracy and speed in processing large arrays of sparse data. | The possibility of using the algorithm in large-scale systems to solve applied problems of user data processing needs to be covered more. |

We analyzed the most recent work and determined that the problems of flexible optimization of big data in industrial systems and selection of the best operating parameters of recommender systems is still relevant. In this work, we propose to deepen the research using data on different numbers of users when providing recommendations. Based on research, it has been determined that the results may vary depending on the volume and characteristics of the data; therefore, it is advisable to choose the most optimal parameters of the Funk SVD algorithm for specific industrial systems. It is also proposed to exchange effective parameters for the work of the recommendation system among several users for faster and more efficient data processing.

### 3. Features of the Smart Manufacture Systems

The development of digital technologies has made it possible to transform various industries and digitize processes that were previously performed only manually. The fourth industrial revolution (Industry 4.0) envisages the operation of production in the form of a flexible and scalable intelligent system that can automatically respond to events and learn on its own. Smart manufacturing allows for the integration of new functions into the operation of the industrial system. Due to the decentralized architecture, a significant number of smart end and edge devices are used. Sharing the load among them improves data reliability and privacy. The hierarchy of the smart manufacturing system is shown in Figure 1.

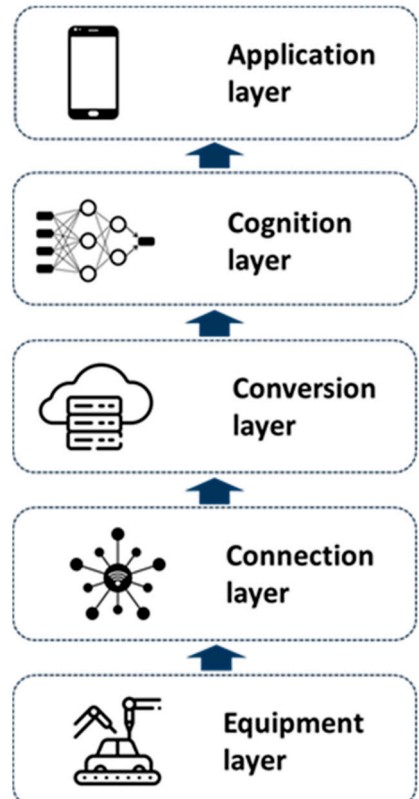

**Figure 1.** Hierarchy of the smart manufacturing system.

According to Figure 1, the devices of the industrial system perform the direct functions of producing goods. With the help of communication between nodes, data exchange, and analysis, the system as a whole is trained to provide services to users. Constant monitoring of the system condition determines the most problematic links in the production process. Analysis of the behavior of users concerning the product allows for the establishment of the most relevant ones and modification or stopping the production of unprofitable ones. The concept of Industry 4.0 involves the transformation of centralized industrial systems into flexible and independent ones. To achieve higher efficiency, all stages of production are considered as a set of interrelated processes, from the production of goods to interaction with users. Integration of the latest technologies such as machine learning, artificial intelligence, cloud computing, and The Internet of Things transforms industrial systems and enables the provision of new services. As a significant number of end devices are used to analyze the state of the system, big data are constantly processed, so quality analytics are critical for the efficiency of production.

Industrial systems are now aimed not only at the manufacture of certain products but also at delivery to the point of sale, interaction with the user, analysis of their evaluations, etc. To a large extent, this approach allows for the adjustment of the production volumes

for certain products, develops new technologies, and offers services that are most likely to be in demand. In the conditions of constant competition in the market and changes in the needs of users, the processing of statistics on the manufacturing operation reduces the probability of producing irrelevant goods and investing in projects that do not find feedback from users. Feedback can be explicit or implicit:

- Explicit reviews are ratings that users leave on their own;
- Collecting data from various sources, such as websites, social networks, and search engines, allows us to receive implicit feedback that indicates how interested users are in certain products or services.

As the amount of data to be processed is constantly increasing, recommendation systems are used that determine the relationships between users and production services. Examples of work are recommendations of videos on YouTube, movies on Netflix, applications in the Play Market, etc. In this way, users receive recommendations for new products or services that are most likely to interest them. According to statistics, a significant part of downloads in the Play Market was made due to recommendations. Recommendation systems facilitate the process of choosing products and finding the most suitable ones for individual categories of users. It is also an opportunity for RS manufacturers to offer their products more effectively.

The process of searching for the most suitable products among users takes place in several stages:

- Selection among the set of products most interesting for the user;
- Evaluation of selected goods;
- Sorting according to rating;
- Providing recommendations to users;
- Correction of results.

The search for the most suitable products for users can be carried out in several ways; the most common are product-based and collaborative filtration. Product-based filtering finds new products that are most similar to those previously liked by an individual user. At the same time, data from other users are not taken into account. For example, if the users prefer one genre of movies, then similar ones they might like are selected. The advantages of content-based filtration are:

- Consideration of user characteristics when forming recommendations;
- It is necessary to process the data of only one user.

The disadvantages of this approach are:

- The impossibility of product recommendations that go beyond usual preferences;
- A significant expenditure of resources.

Another approach is collaborative data filtering, in which a certain degree of similarity between users is first determined. Then, products that have already been positively evaluated by one or more of them are recommended among similar users. With this approach, we can expand the list of recommended services. The advantages of collaborative filtering are:

- Even in the absence of data on the user's preferences, recommendations can be provided;
- More diverse offers of goods;
- Ease of implementation.

Disadvantages of the collaborative filtering method:

- Cold-start problem that occurs when new elements in the system are not processed immediately and do not fall into the recommendations for the corresponding category. Solved by automatically adding a single item to a category without retraining the entire system;
- Difficulties in taking into account additional factors that may affect the accuracy of recommendations. For example, when choosing a product, such factors can be the

user's age, country of residence, etc. To improve the performance of collaborative filtering systems, additional parameter calculation functions are introduced.

To improve the effectiveness of recommendations, data are shared about the results of different users. The general architecture of the recommendation system is shown in Figure 2.

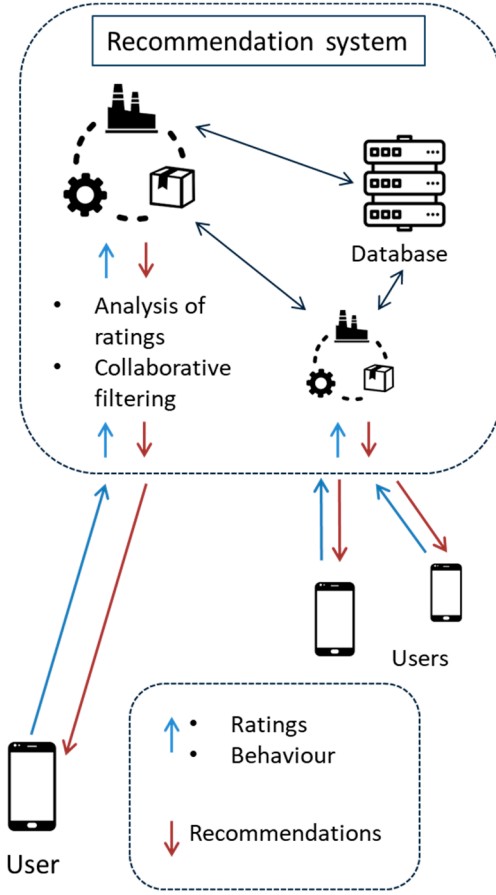

**Figure 2.** Recommendation system architecture.

Determining the most interesting products for users helps to adjust manufacturing parameters. Personalized recommendations that users can receive from social networks, e-mail, and websites based on previous actions are known to make it easier to find the most suitable service among many services. An important challenge for recommendation systems is the security and privacy of user information. The use of data from personal devices must be protected from third-party interference and modification. The possibility of incorrect ratings from some users as a result of a mistake or the intervention of intruders should also be considered. To solve the problem, the authentication of users and checking the data coming from them should be constantly monitored.

Based on the ratings of users in the industrial system, it is possible to establish the prospects of manufacturing a certain type of product or providing certain services. If there is a decline in interest in the product, it can be modified or replaced with a new one that better meets the needs of consumers.

## 4. Data Processing in Recommendation Systems

Special algorithms that analyze statistical regularities of information are used for the operation of algorithms for finding regularities among data. The data are in the form of tables for easier perception by algorithms. One of the most commonly used is the SVD algorithm. Due to this, it is possible to decompose the initial data matrix into a product of three

derivative matrices, in which a certain number of rows and columns that are not necessary for further analysis can then be deleted. Due to the relative simplicity of implementation and accuracy of operation, the SVD algorithm is widely used in recommendation systems of various types and has repeatedly proven its effectiveness.

The SVD algorithm involves the decomposition of the data matrix $M$, containing $n$ rows and $m$ columns as a product of three submatrices:

$$M = U \times S \times V^T \tag{1}$$

where $U$—matrix with dimension $(n, n)$, left singular vectors of matrix $M$; $V^T$—matrix with dimension $(m, m)$, right singular vectors of matrix $M$; $S$—matrix with dimension $(n, m)$, which shows the relationships between matrices $U$ and $V$.

The matrices $U$ and $V$ are orthogonal, and equalities are correct:

$$U \times U^T = I \tag{2}$$

$$V \times V^T = I \tag{3}$$

where $I$—the identity matrix.

To find the columns of the $U$ and $V$ matrices, the eigenvectors and eigenvalues of the products $M \times M^T$ and $M^T \times M$ are calculated in accordance. It is considered that the eigenvalue for a certain matrix of matrix $P$ is the some scalar $\theta$, and the eigenvector $\alpha$, if the condition is fulfilled:

$$P \times \alpha = \theta \times \alpha \tag{4}$$

Then

$$M \times M^T = P \tag{5}$$

We can calculate

$$(P - \theta I)\alpha = 0 \tag{6}$$

Representing Equation (6) as a quadratic one, it is possible to obtain its two roots, that is, the eigenvalues $\theta_1$ and $\theta_2$. Then, using the found eigenvalues, the eigenvectors of the matrix $M$ can be calculated.

Matrix $S$ contains data only on the main diagonal, arranged in descending order. Accordingly, these elements (singular numbers) reflect the influence of various factors on the relationship between data. By choosing only the $k$ largest singular numbers and discarding the other rows and columns, it is possible to optimize the matrix $S$. In addition, we are able to discard the extra rows and columns in matrices $U$ and $V$. After multiplying the new matrices, we get a new matrix $M_1$ with dimension $(k, k)$ and repeat the main characteristics of the input, but it is more compact and easier for further processing.

$$M_1 = U_1 \times S_1 \times V_1^T \tag{7}$$

However, for systems that cannot always collect all data, part of the table is incomplete, that is, such a matrix is sparse. To optimize calculations, it is impractical to use the SVD algorithm, which decomposes the initial matrix by three and then reduces it. Instead, the Funk SVD algorithm is used, which immediately decomposes the matrix into the product of two smaller ones. Figure 3 shows the differences between SVD and Funk SVD methods. For calculating the Funk SVD decomposition, two matrices of the same size as input are randomly generated. Then, the difference between the original matrix and their product is iteratively reduced using the gradient descent algorithm. As a result, we get matrices $U$ and $V$. To speed up the calculations, we used the Python TensorFlow library and the Adam optimization method [29]. The parameter $d$ must be smaller than the size of the input matrix and is selected by decomposing the data into harmonics. In this way, the weight of each column and row is determined for the formation of recommendations. If, during the

operation of Funk SVD, it turns out that the effectiveness of providing recommendations is insufficient, parameter $d$ is recalculated, or optimal values are obtained from other systems.

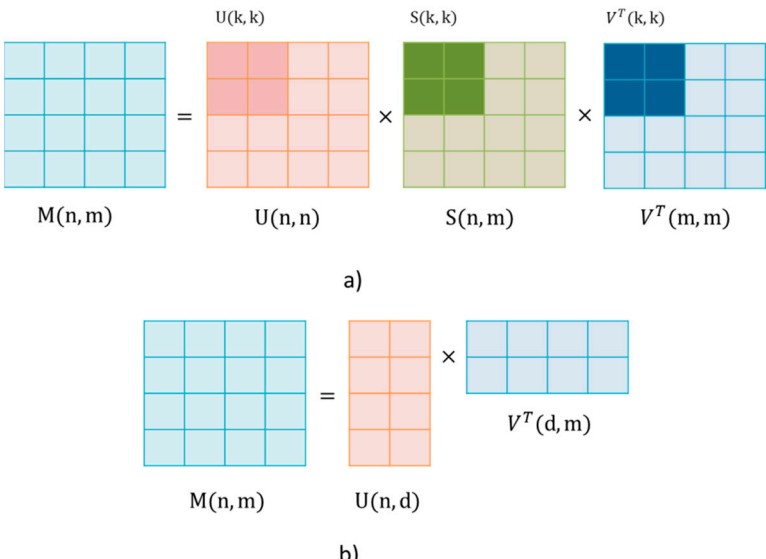

**Figure 3.** The principle of operation of the SVD algorithm (**a**) and Funk SVD (**b**).

As we can see from Figure 3, the singular data decomposition algorithm requires significantly more computing resources and memory to store additional matrices. In contrast, Funk SVD is simpler and computationally less expensive. An additional advantage of using Funk SVD in large-scale recommendation systems is that they often require handling very large sparse matrices. For example, if there is no rating for a certain product, the corresponding matrix cell will be empty. With a high degree of the sparseness of recommendation matrices, a situation may arise when large tables are stored and processed in which the required information occupies an insignificant part. In this case, Funk SVD allows for not decomposing large matrices and the product of the other three before optimization but will immediately choose the required degree of compression. The advantage of this approach is to improve the speed of data processing and reduce the consumption of resources for their storage. We modeled the specified algorithm's work using the Python programming language and the TensorFlow, Pandas, PySpark, NumPy libraries. We used an open database of the Movies Dataset, containing metadata on over 45,000 movies, for modeling. This database contained 26 million ratings from over 270,000 users. Calculations were performed on an Intel(R) Core (TM) i5-4300U CPU @ 1.90 GHz 2.49 GHz in the PyCharm software environment. The duration of calculations for the SVD and Funk SVD algorithms are shown in Figure 4. Here, $k$ and $d$ are the value of the initial matrix, which is further processed (for the SVD and Funk SVD algorithms, accordingly).

As we can see from Figure 4, the Funk SVD algorithm works faster than the usual SVD. With the same level of data compression, the execution time for Funk SVD is shorter on average by 20% compared to the SVD algorithm. Due to its simplicity, it is possible to process large amounts of information more efficiently. This is an important problem in large-scale recommendation systems, where it is necessary to quickly provide solutions to users. Now let us compare the accuracy of restoration of initial data after decomposition for SVD and Funk SVD algorithms (Figure 5). This analysis shows to what extent the matrix after decomposition corresponds to the characteristics of the initial one.

From Figure 5, we can see that Funk SVD is not inferior in accuracy to SVD, so it does not pose a problem for its use. Instead, better calculation speed and ease of implementation reduces the load on computing devices and increases the level of service provision to users. Especially when processing sparse data, the Funk SVD method allows one to perform fewer stages of processing and obtain a quick result.

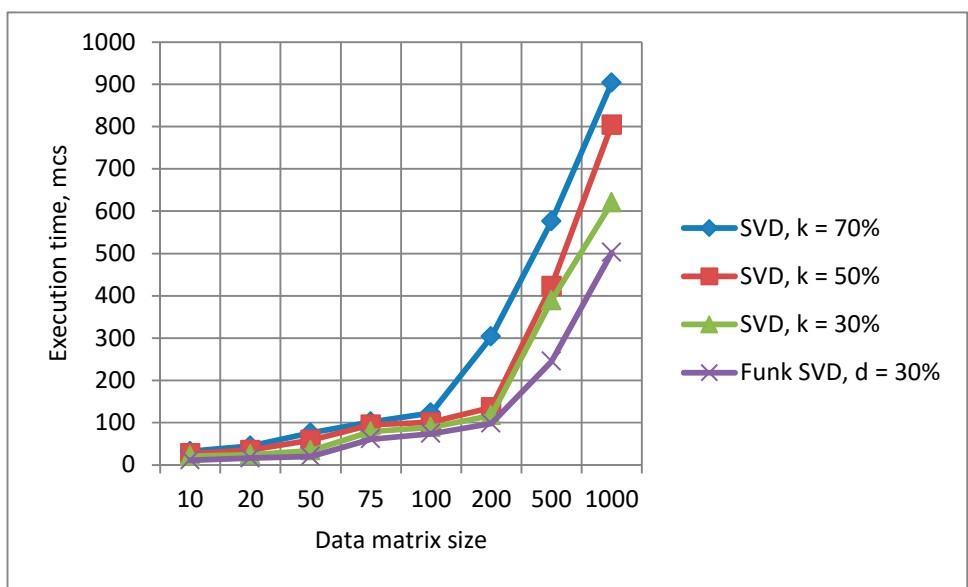

**Figure 4.** Execution time comparison for the SVD and Funk SVD algorithms.

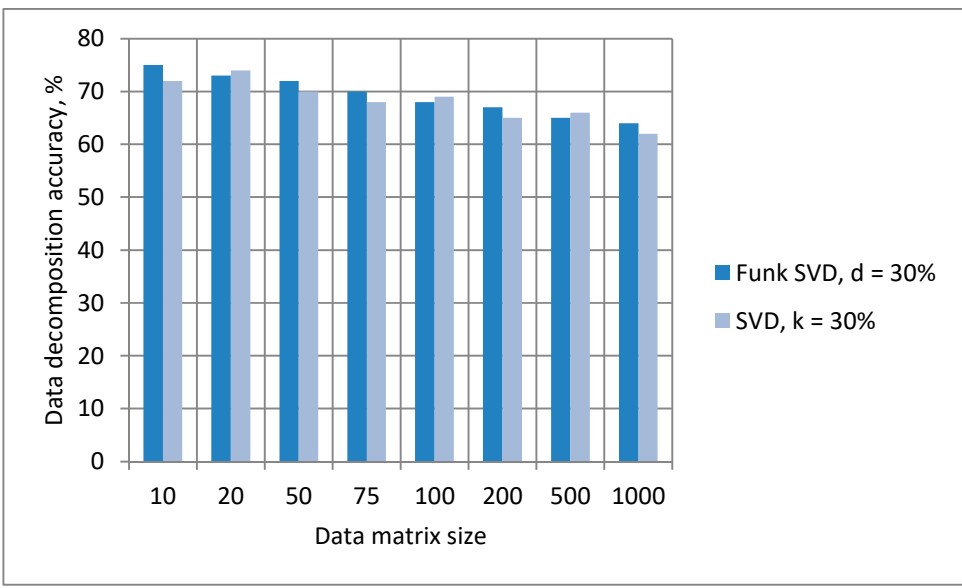

**Figure 5.** Data decomposition accuracy comparison for the SVD and Funk SVD algorithms.

For determining the necessary and optimal amount of data that should be used in further research, it is possible to divide them into harmonics. Such a method shows the amount of information that should be needed. Therefore, only part of the data is needed to make a recommendation. A demonstration of this approach is shown in Figure 6.

According to Figure 6, it can be understood that in the data matrix with dimensions of [1000, 1000], only 50 columns and rows contain important information. Thus, we can choose the necessary compression parameter for the Funk SVD decomposition. The relationship between the percentage of data reduction and the accuracy of restoring the initial data after decomposition for the Funk SVD algorithm is shown in Figure 7.

Figure 7 shows that even when discarding a significant part of the data, this does not significantly affect the accuracy of the calculations. Let us take a closer look at data processing and the formation of recommendations by the Funk SVD algorithm. As noted above, the input matrix is decomposed into the product of two:

$$M = U \times V^T \tag{8}$$

The input matrix $M$ has the dimension of $n$ rows and $m$ columns, i.e., $(n, m)$, $U$ and $V^T$, respectively, $(n, d)$ and $(d, m)$, $d < n$ and $m$.

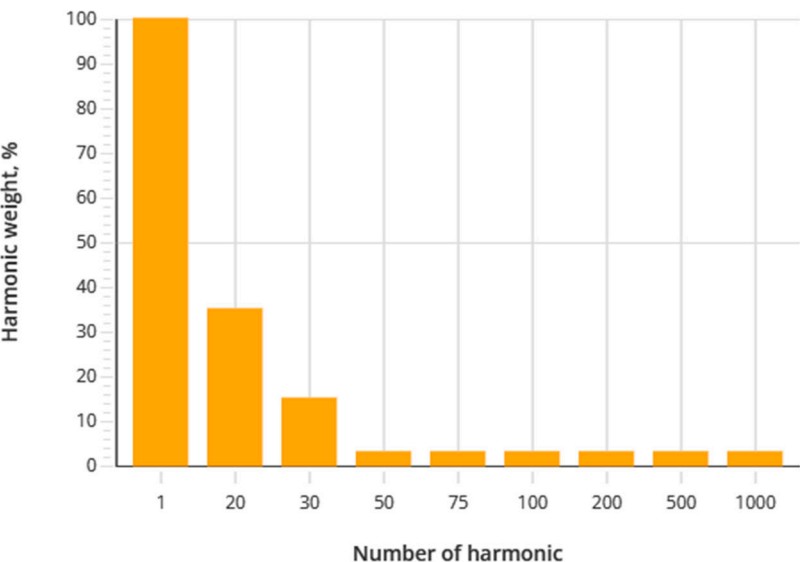

**Figure 6.** Data harmonic weight dependence on its number.

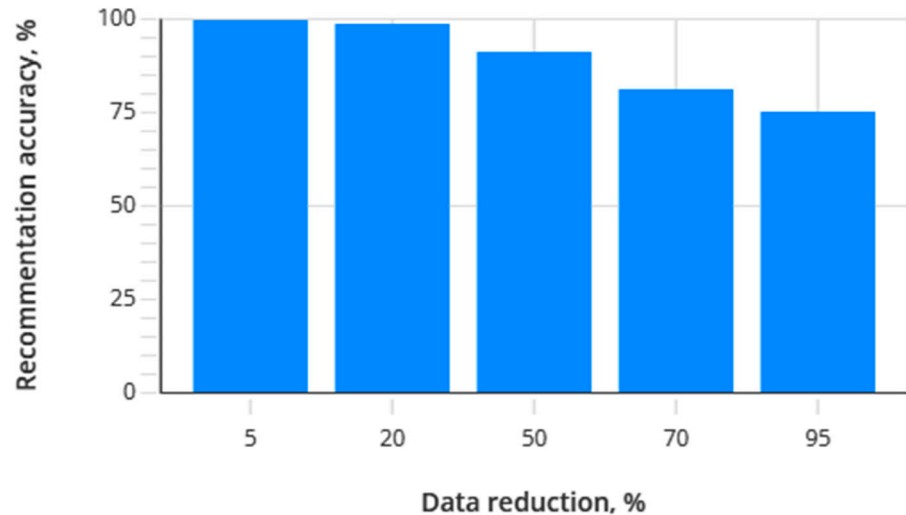

**Figure 7.** Recommendation accuracy dependence on data reduction.

After the schedule, we get new matrices that are responsible for the behavior of users and products. For determining the value of a cell in the input matrix $M$ with index $(i,j)$, we can use the formula

$$m_{i,j\_predicted} = \sum_{g,h=d} \left( u_{g,i} \times v_{j,h} \right) \tag{9}$$

For calculating the error between the actual and expected values of the matrix:

$$k_{err} = \left( m_{i,j} - m_{i,j\_predicted} \right)^2 \tag{10}$$

We can update the values of the $U$ and $V$ matrices:

$$u_{i,j\_new} = u_{i,j} + 2\theta \left( m_{i,j} + m_{i,j_{predicted}} \right) \times v_{i,j} \tag{11}$$

$$v_{i,j\_new} = v_{i,j} + 2\theta \left( m_{i,j} + m_{i,j_{predicted}} \right) \times u_{i,j} \tag{12}$$

The new value for matrix $M$:

$$m_{i,j\_predicted\_new} = \sum_{g,h=d} (u_{g,i\_new} \times v_{j,h\_new}) \tag{13}$$

The work proposes to use not all values of the matrix $U$, which is responsible for user characteristics, but only one or a few, thus improving the speed of providing recommendations:

$$m_{i,j\_predicted} = u_i \times \sum_{h=d} v_{j,h} \tag{14}$$

The relationship between the duration of calculations and the number of users taken into account when forming recommendations is shown in Figure 8.

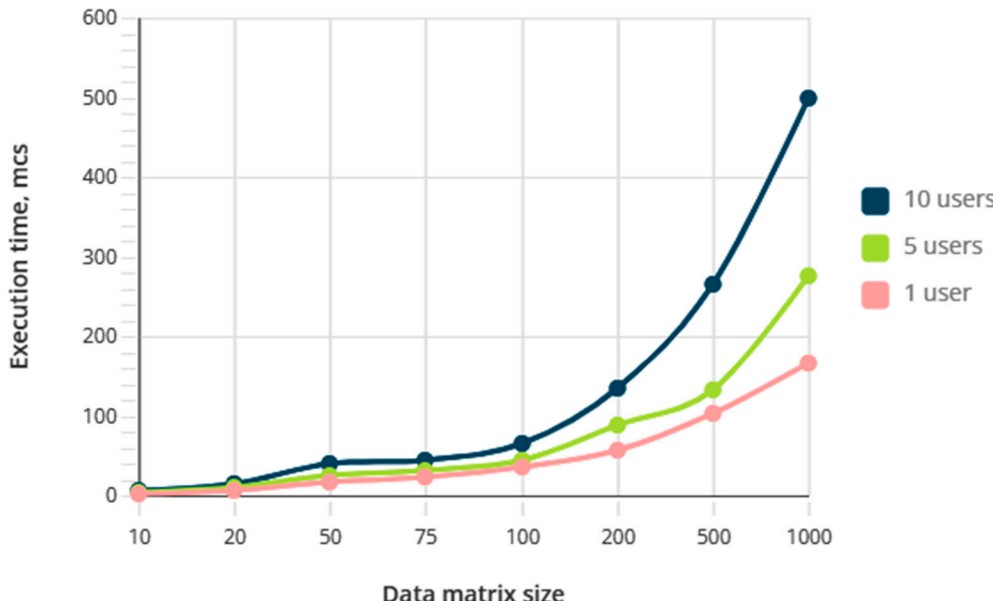

**Figure 8.** Execution time for the Funk SVD algorithm depending on the number of users involved in providing recommendations.

Figure 8 shows that the execution time for Funk SVD using information about one user to provide recommendations is shorter on average by 70% compared to Funk SVD using information about ten users and on average by 50% compared to using information about five users.

The dependence between the duration of calculations and the number of users taken into account when forming recommendations is shown in Figure 9.

Considering the structure of the recommendation system using the Funk SVD algorithm, Figure 10a shows the classic Funk SVD, which uses data about all users. Figure 10b shows a modified approach when there are a limited number of users.

For a faster selection of Funk SVD operation parameters, it is also suggested to exchange the found optimal values between nodes. For example, the optimal parameter $d$, which is responsible for the degree of data compression, can be calculated on one node and transmitted to others. The dependence of the duration of calculations on the number of nodes in groups n that exchange values for data optimization is shown in Figure 11.

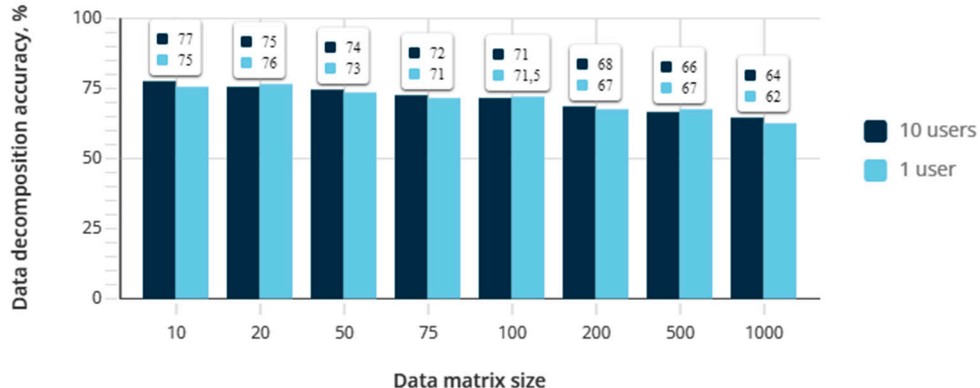

**Figure 9.** Data decomposition accuracy for the Funk SVD algorithm depends on the number of users involved in providing recommendations.

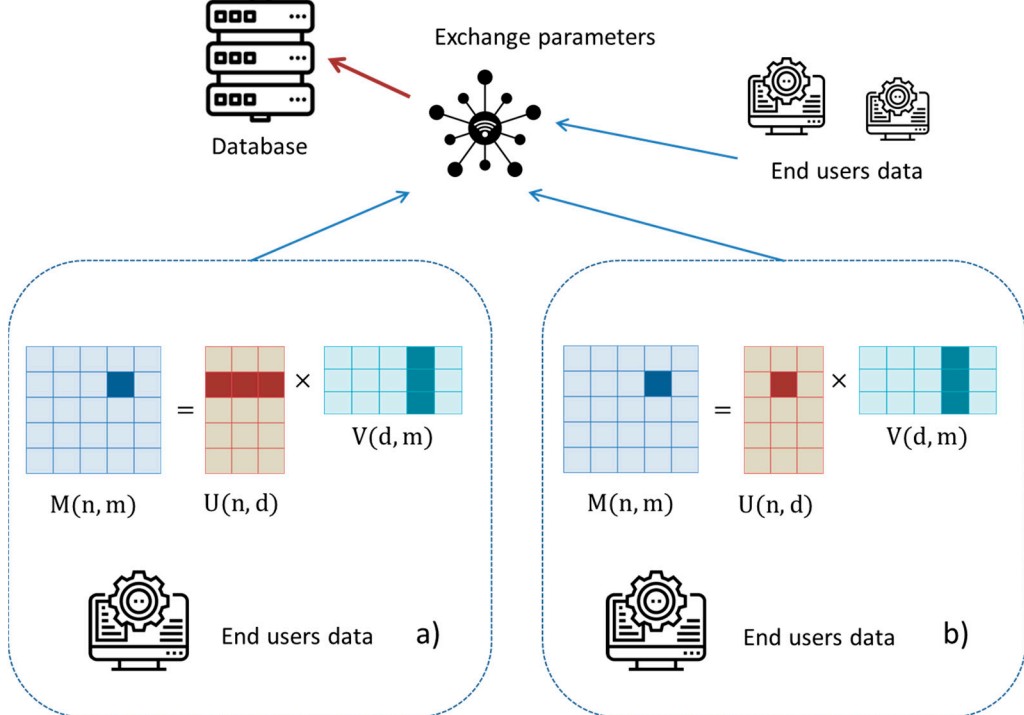

**Figure 10.** Recommendation system using Funk SVD (**a**) and modified Funk SVD (**b**) algorithms.

As we can see from Figure 10, by reducing the number of calculations for each node, the duration of information processing can be reduced. The execution time is shorter on average by 75% for using ten nodes exchanging the optimal decomposition parameter compared to the calculation on one node without exchanging data.

The use of Hadoop and Apache Spark technologies was also compared to perform the main block of calculations. Hadoop technology allows for distributed computing by sending tasks to multiple devices. At the same time, the data are recorded in the permanent memory of the devices. Apache Spark technology allows us to speed up calculations, as it records information in RAM (Figure 12).

As we can see from Figure 12, if it is necessary to perform faster calculations for a large amount of data, Spark technology is more acceptable, because the execution time is shorter on average by 20% compared to the Funk SVD calculation using Hadoop. At the same time, it should be noted that Spark requires a large amount of RAM, which is an additional cost for the system.

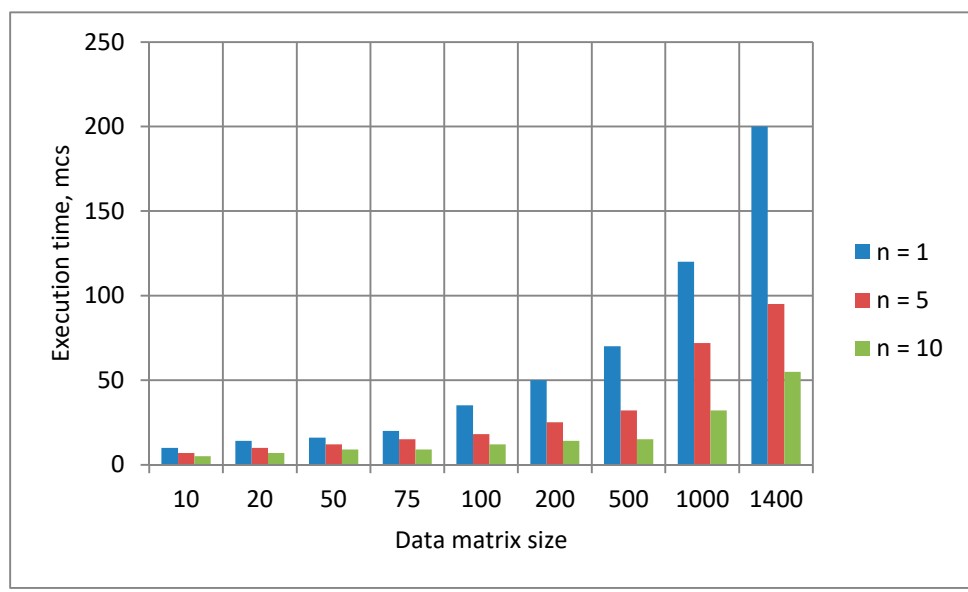

**Figure 11.** Dependence of execution time for the Funk SVD on the number of nodes exchanging the optimal decomposition parameter.

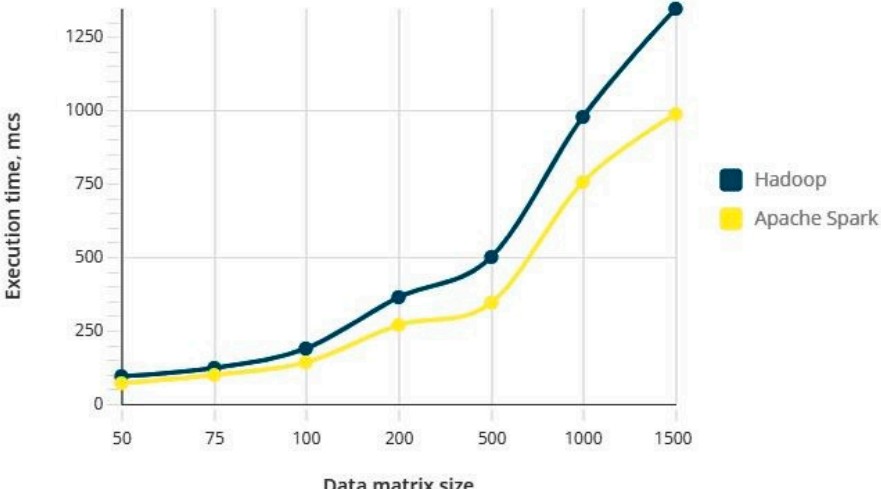

**Figure 12.** Computation speed comparison of Hadoop and Spark's for Funk SVD calculation.

From the results of the conducted research, it can be briefly summarized that for the analysis of large data about users of industrial production, it is necessary to use recommendation systems. The SVD algorithm allows us to process information and find relationships in it. Funk SVD is more efficient for sparse data because it runs faster on average by 20% and requires fewer resources. The proposed modification of Funk SVD, which uses fewer data about users when calculating recommendations, showed high accuracy on the used dataset, but with a better speed of information processing, on average by 50–70%. The proposed method is promising for further research on data processing in recommendation systems and, in particular, smart manufacturing systems and IIoT. Based on the proposed research, an architecture of the industrial recommendation system is proposed (Figure 13), which uses cloud-edge devices that collect and process data from various end users.

According to Figure 12, the use of edge devices allows for collecting data about the work of various types of users. Edge devices receive sparse data matrices that are optimized, and recommendations to users are formed using the Funk SVD algorithm. The management center of the IIoT system and cloud resources are used to aggregate the results, monitor the effectiveness, and adjust the parameters of the recommender system. To summarize,

the proposed architecture of the industrial system using cloud resources and the modified Funk SVD algorithm allows us to quickly process data from various subsystems and use them for effective management and improvement of work quality.

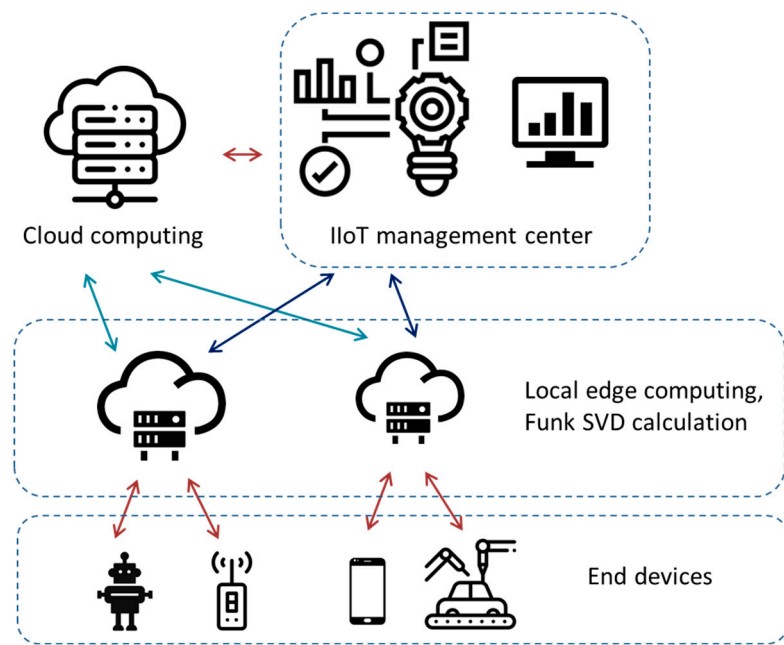

**Figure 13.** The architecture of the industrial recommendation system using Funk SVD.

### 5. Conclusions

The problems of industrial manufacturing data processing to find products that are most suitable for a specific person were examined in the paper. The advantages of using recommendation systems and the Funk SVD algorithm for processing sparse data matrices were determined. A modification of Funk SVD was proposed that requires fewer user data to be used.

It was determined that the proposed Funk SVD calculation method allows for a speed-up of the recommendations process on average by 50–70% depending on the number of users and also requires less involvement of computing resources. At the same time, a fairly high accuracy of data processing was maintained.

For a faster selection of modified Funk SVD operation parameters, it was also proposed to exchange the found optimal values between nodes. As a result of modeling, it was determined that the duration of information processing can be reduced on average by 75% for using ten nodes exchanging the optimal decomposition parameter compared to using one. Using Spark technology for calculating Funk SVD was proposed for reducing the execution time on average by 20% compared to Hadoop. The obtained results show the prospect of further research on the proposed method for data processing in industrial systems. The architecture of the Industrial Internet of Things system, which uses the Funk SVD algorithm for processing user data on edge devices, was proposed. The result is collected on control devices and cloud resources for analysis and effective system management. In future studies, we plan to continue the study of the modified Funk SVD to create algorithms for the automatic selection of optimal calculation parameters for a specific problem. We are working to create a recommender system architecture that works effectively under changing workloads.

**Author Contributions:** All authors contributed to the study conception and design. Methodology, M.B.; software, O.H.-B.; validation, H.B., I.I. and O.H.-B.; formal analysis, H.B.; investigation, M.B.; resources, O.H.-B.; data curation, I.I.; writing—original draft preparation, M.B.; writing—review

and editing, M.B., O.H.-B.; visualization, I.I.; supervision, O.H.-B.; project administration, M.B. All authors have read and agreed to the published version of the manuscript.

**Funding:** This research was supported by the project No. 0120U102201 "Development the methods and unified software-hardware means for the deployment of the energy efficient intent-based multi-purpose information and communication networks" and by the project in Ukraine "Development the innovative methods and models of designing the industry-oriented information and communication systems for upgrading the digital industrial infrastructures".

**Institutional Review Board Statement:** Not applicable.

**Informed Consent Statement:** Not applicable.

**Data Availability Statement:** Not applicable.

**Conflicts of Interest:** The authors declare no conflict of interest.

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
