# Peer review of "Data Optimization for Industrial IoT-Based Recommendation Systems"

_electronics, doi:10.3390/electronics12010033_

Round 1

Reviewer 1 Report

This is a well-written manuscript on data processing and optimization methods for IIoT based recommendation systems. It will interest a reasonably broad readership of Electronics. I have the following comments that the authors should address before final acceptance:

1.     Equation 7 seems missing a transpose sign for matrix V1.

2.     The authors should double-check the spelling and typos in the manuscript. For example, in the caption of Figure 5, “and” is miss-spelled as “abd”; Funk SVD is spelled as both “FunkSVD” and “Funk SVD” in the text, etc.

3.     When calculating the error between the actual and expected values of the matrix in Equation 10, it should be a minus sign instead of a plus sign.

4.     For all the experiments that the authors did, I think it is necessary to mention how many times the experiments were performed, and add error bars to the figures, since the authors are trying to quantify how better their optimizations are.

Author Response

First of all, we are very grateful for the time given by the reviewer to review the submitted manuscript (Data Optimization for Industrial IoT-Based Recommendation Systems) and also for the inspiring comments. We have made every effort to eliminate all the indicated flaws and inconsistencies and hope that as a result, the quality of our manuscript has improved further. We have included this response letter with the revised submission of the manuscript in order to answer the reviewers’ questions, clarify any confusing parts of our submission and provide the list of changes that we have made in order to reflect the comments of the reviewers. All changes made to the revised manuscript are highlighted in red in the review mode of word files.

Response to Reviewer 1 Comments

Comments and Suggestions for Authors

This is a well-written manuscript on data processing and optimization methods for IIoT based recommendation systems. It will interest a reasonably broad readership of Electronics. I have the following comments that the authors should address before final acceptance:

Point 1:Equation 7 seems missing a transpose sign for matrix V1.

Response 1: Thank you for the remark. We added a transpose sign for matrix V1

Point 2:The authors should double-check the spelling and typos in the manuscript. For example, in the caption of Figure 5, “and” is miss-spelled as “abd”; Funk SVD is spelled as both “FunkSVD” and “Funk SVD” in the text, etc.

Response 2: We corrected the text

Point 3:When calculating the error between the actual and expected values of the matrix in Equation 10, it should be a minus sign instead of a plus sign.

Response 3: We corrected equation 10

Point 4:For all the experiments that the authors did, I think it is necessary to mention how many times the experiments were performed, and add error bars to the figures, since the authors are trying to quantify how better their optimizations are.

Response 4: We performed one experiment for each data set of different volume.

Reviewer 2 Report

The manuscript is well written. My comments are given below.

1) The abstract is very long. It must be reduced according to the journal format.

2) The motivation of this study is missing. I strongly recommend you add in the introduction section.

3) The contributions and advantages of this study are missing. Please add and highlight using bullets.

4)In a related world please make a comparison table and discuss five or six most recent studies along with advantages and disadvantages.

5) The conclusion section must be shortened and also discuss the future work of your study.

6) Please mention the details of datasets and the simulation tool that you have used in the experimentation.

Author Response

First of all, we are very grateful for the time given by the reviewer to review the submitted manuscript (Data Optimization for Industrial IoT-Based Recommendation Systems) and also for the inspiring comments. We have made every effort to eliminate all the indicated flaws and inconsistencies and hope that as a result, the quality of our manuscript has improved further. We have included this response letter with the revised submission of the manuscript in order to answer the reviewers’ questions, clarify any confusing parts of our submission and provide the list of changes that we have made in order to reflect the comments of the reviewers. All changes made to the revised manuscript are highlighted in red in the review mode of word files.

Response to Reviewer 2 Comments

Comments and Suggestions for Authors

The manuscript is well written. My comments are given below.

Point 1:The abstract is very long. It must be reduced according to the journal format.

Response 1: Corrected the abstract:

The most common problems that arise when working with big data for intelligent production are analyzed in the article. The work of recommendation systems for finding the most relevant users information was considered. The features of the Singular-Value Decomposition (SVD) and Funk SVD algorithms for reducing the dimensionality of data and providing quick recommendations were determined. An improvement of the Funk SVD algorithm using a smaller required amount of user data for analysis was proposed. According to the results of the experiments, the proposed modification improves the speed of data processing on average by 50-70% depending on the number of users and allows spending less computing resources. As follows, recommendations to users are provided in a shorter period and are more relevant. The faster calculation of modified Funk SVD to exchange the optimal parameters between nodes was proposed. That execution time can be reduced on average by 75% for using ten nodes exchanging the optimal decomposition parameter compared to using one was determined. Using Spark technology for faster calculation on average by 20% compared to Hadoop was proposed. The architecture of the IIoT system was proposed, which uses a modified Funk SVD algorithm to optimize data on edge devices and monitor the effectiveness of providing recommendations using control centers and cloud resources.

Point 2:The motivation of this study is missing. I strongly recommend you add in the introduction section.

Response 2: Added a text:

The problem of processing big data and providing relevant recommendations to users in industrial systems is still relevant. We analyzed the Funk SVD algorithm and found it is effectively used for processing sparse big data matrices. However, no studies show its effectiveness when working in conditions where the load on the system can change dynamically, so the optimal parameters for big data processing should be selected accordingly. We analyzed existing solutions and proposed modifications of Funk SVD, which allowed us to speed up the calculation process by discarding redundant data about users. Also, with high requirements for the accuracy and speed of data processing, we proposed the architecture of a recommender system with exchanging optimal calculation parameters that help form recommendations faster.

Point 3:The contributions and advantages of this study are missing. Please add and highlight using bullets.

Response 3: We added the main contributions and advantages of this study to Introduction section:

  • the main research in the field of processing big data about users was analyzed, and the advantages of the Funk SVD algorithm were revealed;
  • a modified Funk SVD was proposed, which works more efficiently in systems with high requirements for calculation speed because when forming recommendations, data is not used about all users, but only a part, thus discarding information that does not significantly affect the accuracy of calculations;
  • the second modification of Funk SVD was proposed, which allows exchanging optimal parameters between different nodes, that affect the effectiveness of providing recommendations;
  • for calculating Funk SVD in the systems of the Industrial Internet of Things, the Spark distributed computing technology was proposed, and the improvement of the speed of calculations, especially of large data sets, was determined;
  • based on the research, the architecture of recommender systems in IIoT working with variable work requirements was proposed. In this way, it is possible to flexibly apply the proposed modifications to achieve the maximum efficiency of user data processing.

Point 4:In a related world please make a comparison table and discuss five or six most recent studies along with advantages and disadvantages.

Response 4: Added a comparison table and study motivation:

Table 1 shows a comparison of the most recent studies along with advantages and disadvantages in the field of big data processing in recommender systems.

Table 1. Сomparison of the most recent studies along of big data processing in recommender systems.

Study

Advantages

Disadvantages.

[18]

Authors proposed an integrated online and offline real-time recommendation service providing a variety of analysis methods to realize data mining.

The possibilities of applying the proposed method and optimization of big data should be covered more.

[20]

Proposed a recommendation system that improves the operation of the traffic management system by learning based on human behavior

No methods are given to optimize big data obtained from different users and adaptation to different workloads.

[25]

Investigated the advantages of the Funk SVD algorithm for working with sparse data and proposed a new recommendation algorithm that combines Funk-SVD and K-means

Features of operation for variable load systems are not given

[26]

Proposed to improve and simplify Funk SVD, using only one user's data when forming recommendations, proving that the accuracy of calculations does not deteriorate.

No comparison is given on how the effectiveness of providing recommendations changes when processing data from different numbers of users

[27]

Developed a recommendation system based on machine learning that can be applied to almost any application

There are no defined problems of different volumes data  processing  and service requirements

[34]

Offered a parallel GPU-based Funk SVD algorithm, demonstrating high accuracy and speed in processing large arrays of sparse data.

The possibility of using the algorithm in large-scale systems to solve applied problems of user data processing needs to be covered more.

We analyzed the most recent works and determined that the problem of flexible optimization of big data in industrial systems and selecting the best operating parameters of recommender systems is still relevant. In this work, we propose to deepen the research, using data on different numbers of users when providing recommendations. Based on research, it is determined that the results may vary depending on the volume and characteristics of the data; therefore, it is advisable to choose the most optimal parameters of the Funk SVD algorithm for specific industrial systems. It is also proposed to exchange effective parameters for the work of the recommendation system between several users for faster and more efficient data processing.

Point 5:The conclusion section must be shortened and also discuss the future work of your study.

Response 5: Shortened the conclusion section and added and discuss the future work

The problems of industrial manufacturing data processing to find products that are most suitable for a specific person were examined in the paper. The advantages of using recommendation systems and the Funk SVD algorithm for processing sparse data matrices were determined. A modification of Funk SVD was proposed that requires fewer user data to be used. A study of the operation of the modified method and a comparison with the usual one were carried out. It was determined that the proposed Funk SVD calculation method allows for a speed-up of the providing recommendations process on average by 50-70% depending on the number of users and also requires less involvement of computing resources. At the same time, a fairly high accuracy of data processing was maintained.

For a faster selection of modified Funk SVD operation parameters, it was also proposed to exchange the found optimal values between nodes. As a result of modeling was determined that the duration of information processing can be reduced on average by 75% for using 10 nodes exchanging the optimal decomposition parameter compared to using one. Using Spark technology for calculating Funk SVD was proposed for reducing the execution time on average by 20% compared to Hadoop. The obtained results show the prospect of further research on the proposed method for data processing in industrial systems. The architecture of the Industrial Internet of Things system, which uses the Funk SVD algorithm for processing user data on edge devices, was proposed. The result is collected on control devices and cloud resources for analysis and effective system management. In future studies, we plan to continue the study of the modified Funk SVD to create algorithms for the automatic selection of optimal calculation parameters for a specific problem. We are working to create a recommender system architecture that works effectively under changing workloads.

Point 6:Please mention the details of datasets and the simulation tool that you have used in the experimentation.

Response 6: Added a following text:

The advantage of this approach is to improve the speed of data processing and reduce the consumption of resources for their storage. We modeled the specified algorithm's work using the Python programming language and the TensorFlow, Pandas, PySpark, NumPy libraryies.  We used an open database of the Movies Dataset, containing metadata on over 45,000 movies, for modeling. 26 million ratings from over 270,000 users. Calculations were performed on an Intel(R) Core (TM) i5-4300U CPU @ 1.90GHz 2.49 GHz. in the PyCharm software environment. The duration of calculations for the SVD and Funk SVD algorithms are shown in Figure 4. Here, k and d are the value of the initial matrix, which is further processed (for the SVD and Funk SVD algorithms, accordingly).

Reviewer 3 Report

Please see the attached comments.

Author Response

First of all, we are very grateful for the time given by the reviewer to review the submitted manuscript (Data Optimization for Industrial IoT-Based Recommendation Systems) and also for the inspiring comments. We have made every effort to eliminate all the indicated flaws and inconsistencies and hope that as a result, the quality of our manuscript has improved further. We have included this response letter with the revised submission of the manuscript in order to answer the reviewers’ questions, clarify any confusing parts of our submission and provide the list of changes that we have made in order to reflect the comments of the reviewers. All changes made to the revised manuscript are highlighted in red in the review mode of word files.

Response to Reviewer 3 Comments

This paper considered the recommendation systems for finding the most relevant users information. Honestly, the authors seem have done a solid work. However, some parts are not well written and clear, and needs to be further clarified. The reviewer has the following concerns:

Point 1:The abstract is not well written. First, the abstract is too long, and the background should be moved to Introduction part. Second, the transitional sentences between background and main work is missing, the authors should briefly introduce the facing and problems or challenges, and then the main work. Third, what is the meaning of “The most critical issues”? Forth, the description of what has been done is not clear.

Response 1: Corrected the abstract:

The most common problems that arise when working with big data for intelligent production are analyzed in the article. The work of recommendation systems for finding the most relevant users information was considered. The features of the Singular-Value Decomposition (SVD) and Funk SVD algorithms for reducing the dimensionality of data and providing quick recommendations were determined. An improvement of the Funk SVD algorithm using a smaller required amount of user data for analysis was proposed. According to the results of the experiments, the proposed modification improves the speed of data processing on average by 50-70% depending on the number of users and allows spending less computing resources. As follows, recommendations to users are provided in a shorter period and are more relevant. The faster calculation of modified Funk SVD to exchange the optimal parameters between nodes was proposed. That execution time can be reduced on average by 75% for using ten nodes exchanging the optimal decomposition parameter compared to using one was determined. Using Spark technology for faster calculation on average by 20% compared to Hadoop was proposed. The architecture of the IIoT system was proposed, which uses a modified Funk SVD algorithm to optimize data on edge devices and monitor the effectiveness of providing recommendations using control centers and cloud resources.

Point 2:The authors should use past tense to introduce the existing works.

Response 2: Used past tense to introduce the existing works

Point 3:One of the critical issues in this paper is the motivations and contributions, the authors only describe what the literatures have done, and then directly move to what this work do. What motivates authors to do this work should be emphasized. Besides, the contributions should be rewritten by listing several points for not only introducing the detailed works, but also the possible application to practical IoT system.

Response 3:

1) We added the motivations of this study to the Introduction section:

The problem of processing big data and providing relevant recommendations to users in industrial systems is still relevant. We analyzed the Funk SVD algorithm and found it is effectively used for processing sparse big data matrices. However, no studies show its effectiveness when working in conditions where the load on the system can change dynamically, so the optimal parameters for big data processing should be selected accordingly. We analyzed existing solutions and proposed modifications of Funk SVD, which allowed us to speed up the calculation process by discarding redundant data about users. Also, with high requirements for the accuracy and speed of data processing, we proposed the architecture of a recommender system with exchanging optimal calculation parameters that help form recommendations faster.

2) We added the main contributions and advantages of this study to the Introduction part:

  • the main research in the field of processing big data about users was analyzed, and the advantages of the Funk SVD algorithm were revealed;
  • a modified Funk SVD was proposed, which works more efficiently in systems with high requirements for calculation speed because when forming recommendations, data is not used about all users, but only a part, thus discarding information that does not significantly affect the accuracy of calculations;
  • the second modification of Funk SVD was proposed, which allows exchanging optimal parameters between different nodes, that affect the effectiveness of providing recommendations;
  • for calculating Funk SVD in the systems of the Industrial Internet of Things, the Spark distributed computing technology was proposed, and the improvement of the speed of calculations, especially of large data sets, was determined;
  • based on the research, the architecture of recommender systems in IIoT working with variable work requirements was proposed. In this way, it is possible to flexibly apply the proposed modifications to achieve the maximum efficiency of user data processing.

Point 4:It is suggested to introduce the following recent works in optimization [R1]-[R3] and IoT [R4] fields to highlight the state-of-the-art of this paper. [R1] “Refracting RIS aided hybrid satellite-terrestrial relay networks: Joint beamforming design and optimization,” IEEE Transactions on Aerospace and Electronic Systems, vol. 58, no. 4, pp. 3717-3724, Aug. 2022. [R2] “SLNR-based secure energy efficient beamforming in multibeam satellite systems,” IEEE Transactions on Aerospace and Electronic Systems, early access, Jul. 2022, doi: 10.1109/TAES.2022.3190238. [R3] “Joint beamforming and power allocation for satellite-terrestrial integrated networks with non-orthogonal multiple access,” IEEE Journal of Selected Topics in Signal Processing, vol. 13, no. 3, pp. 657-670, June 2019. [R4] “Joint beamforming design for secure RIS-assisted IoT networks,” IEEE Internet of Things Journal, early access, Sep. 2022, doi: 10.1109/JIOT.2022.3210115.

Response 4: We have added the following recent works that you recommend, they should definitely be mentioned in our work because they make a certain contribution to data optimization

  1. Z. Lin et al., "Refracting RIS-Aided Hybrid Satellite-Terrestrial Relay Networks: Joint Beamforming Design and Optimization," in IEEE Transactions on Aerospace and Electronic Systems, vol. 58, no. 4, pp. 3717-3724, Aug. 2022, doi: 10.1109/TAES.2022.3155711.
  2. Z. Lin et al., "SLNR-based Secure Energy Efficient Beamforming in Multibeam Satellite Systems," in IEEE Transactions on Aerospace and Electronic Systems, 2022, doi: 10.1109/TAES.2022.3190238.
  3. Z. Lin, M. Lin, J. -B. Wang, T. de Cola and J. Wang, "Joint Beamforming and Power Allocation for Satellite-Terrestrial Integrated Networks With Non-Orthogonal Multiple Access," in IEEE Journal of Selected Topics in Signal Processing, vol. 13, no. 3, pp. 657-670, June 2019, doi: 10.1109/JSTSP.2019.2899731.
  4. H. Niu et al., "Joint Beamforming Design for Secure RIS-Assisted IoT Networks," in IEEE Internet of Things Journal, 2022, doi: 10.1109/JIOT.2022.3210115.

Introduced the following recent works:

In works [30-32], the authors developed original optimization methods that use the decomposition of singular values and the duality of uplink and downlink channels to optimize the weight vectors in order to form energy-efficient beamforming for a satellite-to-ground relay network. In [33] authors proposed secure communication in IoT networks by jointly optimizing the power allocation factors, beamforming vector, and phase shifts, where the confidential signal is sent by an active refracting reconfigurable intelligent surface-based transmitter, to solve a secrecy energy maximization problem.

Point 5:As shown in Fig. 3(b), how to choose the parameter d in Funk SVD algorithm? And how to operate Funk SVD? The detailed explanation should be provided. From my point of view, the parameter d is closely related to the rank of the original information matrix, if not appropriate, the Funk SVD would fail.

Response 5: We added the following text:

For calculating the Funk SVD decomposition, two matrices of the same size as input are randomly generated. Then the difference between the original matrix and their product is iteratively reduced using the gradient descent algorithm. As a result, we get matrices U and V. To speed up the calculations, we used the Python TensorFlow library and the Adam optimization method [29]. The parameter d must be smaller than the size of the input matrix and is selected by decomposing the data into harmonics. In this way, the weight of each column and row is determined for the formation of recommendations. If during the operation of Funk SVD, it turns out that the effectiveness of providing recommendations is insufficient, parameter d is recalculated, or optimal values are obtained from other systems.

Round 2

Reviewer 2 Report

The authors has incorporated all necessary changes to the revised manuscript.

Reviewer 3 Report

The authors have addressed my concerns, no further comments.